# Fairness Through Independence via Cramér-von Mises Regularization

**Albert Gimó**
Criteo AI Lab
a.gimo@criteo.com

**Mariia Vladimirova**
Criteo AI Lab, Fairplay joint team
m.vladimirova@criteo.com

**Olga Petrova**
Criteo AI Lab
o.petrova@criteo.com

**Federico Pavone**[*]
Theremia
federicopavone@theremia.health

**Reda Chhaibi**
Université Côte d'Azur, CNRS, LJAD
reda.chhaibi@univ-cotedazur.fr

## Abstract

Controlling fairness in machine learning (ML) model outputs is challenging due to complex, unstable and computationally expensive techniques for bias estimation on finite data samples. We propose a simple in-processing method to control group fairness during training by penalizing statistical dependence between model outputs $\hat{Y}$ and a sensitive attribute $S$. Our approach instantiates the Cramér–von Mises (CvM) dependence coefficient $\xi(S, \hat{Y})$ as a bounded, differentiable regularizer that integrates seamlessly with stochastic optimization. The resulting objective $L + \lambda\,\xi(S, \hat{Y})$ positions models along a fairness–utility Pareto frontier through a single multiplier $\lambda$. Our experiments demonstrate the effectiveness of this method for controlling the fairness-utility trade-off in both small fairness-aware and large tabular datasets. In order to control the compromise between fairness metrics and utility metrics, we propose a task-agnostic hyperparameter tuning pipeline and showcase its effectiveness in a large tabular dataset. In practice, we have observed that controlling for CvM leads to lower demographic-parity (DP) scores, providing a tractable and computationally efficient methodology, bridging the gap between policy requirements on DP and a scalable training procedure for ML models.

## 1 Introduction

ML models often inherit biases from historical data (through selection effects, under-representation, or label bias) yielding unreliable outcomes for protected groups [5]. High-profile failures in hiring and criminal justice illustrate how models can encode disparities even without explicit access to sensitive attributes $S$ [4, 12]. This reality has led to growing legal and regulatory pressure for model deployers to demonstrate and control fairness, as seen in the EU AI Act and New York City's bias-audit duties. This makes the ability to control fairness, i.e achieving a target level with minimal utility loss, a critical objective for modern ML.

Controlling fairness under biased data is challenging for two reasons. First, population-level fairness objectives, such as disparities in error rates, are often hard to estimate from finite, biased samples, which can lead to wrongly estimated risks. Second, many existing fairness-inducing methods rely on computationally intensive techniques like constrained optimization with expensive projections [2] or adversarial training [21]. These approaches do not scale to the massive datasets and models now common in modern training regimes, e.g. in vision, language, and recommendation systems. A

---

[*]The work was done during a fellowship at Université Paris Dauphine-PSL.

39th Conference on Neural Information Processing Systems (NeurIPS 2025) Workshop: Reliable ML.

practical solution must therefore (i) account for data bias when estimating fairness-relevant quantities and (ii) integrate with standard stochastic training so it scales with dataset and model size.

**Our approach.** We introduce a simple in-processing method that augments standard training losses $L$ with a bounded and differentiable regularizer based on the CvM dependence coefficient $\xi(S, \hat{Y})$, which measures how conditioning on sensitive attribute $S$ shifts the distribution of predictions $\hat{Y}$ [10, 13]. To make $\xi$ trainable, we leverage differentiable ranking [7] to backpropagate through the rank-based estimator introduced in Chatterjee [10], enabling end-to-end optimization of a regularized objective $L + \lambda\xi(S, \hat{Y})$. We study how $\lambda$ positions models along this trade-off and maintain reliable levels of fairness without excessively compromising the model's utility. We provide an analysis on how to perform this adjustment and report results in both standard fairness (*Adults* [1]) and a larger non-fairness specific (*Weather Forecasting* [24]) datasets. Our **contributions** are three-fold:

1. A CvM-based regularizer that promotes independence between $\hat{Y}$ and $S$ on biased datasets, improving the reliability of predictions for protected groups.

2. A differentiable implementation based on soft-ranking with clear stability/complexity properties, exposing a single trade-off parameter $\lambda$ and a smoothness control $\varepsilon$.

3. A scalable training and tuning protocol on small and large tabular workloads that enables practitioners to adjust the fairness-utility trade-offs when training models.

## 2 Method

While many independence measures exist (see discussion in Appendix A.2), they often suffer from estimation difficulty, gradient instability, or interpretability issues. We introduce and study a CvM–based regularizer that measures and minimizes statistical dependence between model outputs $\hat{Y}$ and sensitive attributes $S$, see Appendix B for more details. The CvM dependence coefficient provides a normalized, interpretable scalar in $[0, 1]$ that equals zero iff independence holds and one iff the target is a measurable function of the selected sensitive attribute. It aggregates the variance of conditional expectations of thresholded outcomes, thereby capturing non-linear dependencies without hand-enumerating slices or thresholds. We adopt a finite-sample estimator for the coefficient that is $O(n \log n)$ via sorting and ranking, and we show how to embed it directly in modern optimizers. The CvM coefficient and its estimator are the following:

$$\xi(X, Y) := \frac{\int \mathrm{Var}\left(\mathbb{E}[\mathbb{1}_{\{Y \geq t\}} \mid X]\right) dF_Y(t)}{\int \mathrm{Var}\left(\mathbb{1}_{\{Y \geq t\}}\right) dF_Y(t)} \quad \text{and} \quad \xi_n(X_n, Y_n) := 1 - \frac{n \sum_{k=1}^{n-1} |r_{i_{k+1}} - r_{i_k}|}{2 \sum_{k=1}^{n} l_k(n - l_k)} . \quad (1)$$

The technical challenge in using this estimation for training deep learning models is that ranking is a discrete operation, making it infeasible for gradient–based optimization. We overcome this by leveraging fast and differentiable soft ranking as projections onto the permutahedron from Blondel et al. [7], yielding order-preserving almost-everywhere differentiable operators with exact Jacobians via isotonic optimization. Plugging these operators into the estimator produces a differentiable CvM penalty term that integrates seamlessly with deep learning frameworks and preserves the statistical relevance of the original coefficient.

Crucially for scalability, the proposed objective is minibatch-friendly. The estimator's behavior is especially well-conditioned when the model outputs are continuous (e.g., regression or probabilities in classification), which we recommend in practice. Under continuity, the sample–based coefficient is stable to small perturbations (as described in Proposition 1), improving optimization under SGD noise and batch shuffling.

**Proposition 1** (Robustness to perturbations). *Let $(X_n, Y_n)$ be $n$ i.i.d. samples from $p(X, Y)$. Let $Y$ be continuous and let $Z_n^1, Z_n^2$ contain $n$ i.i.d. samples from a continuous real-valued noise variable. Define $X^\eta := X + \eta Z^1$ and $Y^\eta := Y + \eta Z^2$. Then, with probability 1,*

$$\lim_{\eta \to 0} \xi_n(X, Y^\eta) = \xi_n(X, Y) ,$$

$$\lim_{\eta \to 0} \mathbb{E}\left[\xi_n(X^\eta, Y)\right] = \lim_{\eta \to 0} \mathbb{E}\left[\xi_n(X^\eta, Y^\eta)\right] = \mathbb{E}\left[\xi_n(X, Y)\right] , \quad (2)$$

*where the expectations are with respect to the perturbation noise and any uniformly random tie-breaking mechanism for the right-most term.*

The method integrates into existing training procedures as a single regularization term, with two practical hyperparameters: (i) the soft-ranking smoothness $\varepsilon$ (controls bias–variance of the gradient signal) and (ii) the multiplier $\lambda$ (controls the utility–fairness trade-off). In the experiments section, we provide usage guidance based on our practical results: prefer $L_2$ regularization over $L_1$ on the CvM penalty to increase robustness; when needed, fine-tune from an unregularized checkpoint, while geometrically increasing $\lambda$ to trace a stable Pareto frontier; and we provide a 3 stage method for hyperparameter optimization. These practices aim at preserving accuracy while steadily reducing dependence on sensitive attributes.

## 3    Experiments

We focus on two datasets: (i) `Adult` [1], a canonical fairness benchmark to sanity-check group metrics. We also stress-test by treating education as a sensitive attribute (highly correlated with income) to probe the fairness–utility frontier; (ii) `Weather` (TabReD, Rubachev et al. [24]), a large tabular regression benchmark with deep-learning baselines chosen to test our method's scalability . For more details on the dataset choice and their limitations, we refer to Appendix A.1.

### 3.1    Adult dataset

**Setup.** We consider binary income prediction with sensitive attributes comprising (i) a weakly correlated attribute (gender) and (ii) a strongly correlated attribute (education). Utility is measured via accuracy/F1-score; fairness via DP/EO and CvM. A detailed per-attribute analysis, pre-processing choices and discussion are referred to Appendix E.

**Results.**  We (i) demonstrate how adding our CvM-based regularizer translates to decreases in group-based fairness metrics such as DP, and (ii) study the role of the penalty form and fine-tuning.

- *Multiplier control.* Increasing $\lambda$ monotonically reduces CvM and typically shrinks DP/EO gaps. For weakly correlated attributes, these improvements incur modest utility loss; for highly correlated attributes, utility drops are sharper, consistent with a steeper trade-off frontier, see Figure 1.

- *Penalty form.* Applying an $L_2$ penalty on the CvM term (Figure 1) yields reduced sensitivity to small coefficient changes and increased robustness when compared to $L_1$ regularization (Figure 9). We adopt the $L_2$ formulation henceforth to improve the control over the adjusting of $\lambda$. We refer to Appendix E for more details.

- *Fine-tuning.* The presented regularization can also be introduced as a fine-tuning method. Experimentally, introducing $\lambda$ post-hoc and ramping it geometrically stabilizes training and yields gradual fairness gains with limited utility degradation.

### 3.2    Weather forecasting dataset

**Setup.** We study large-scale temperature prediction using the *Weather Forecasting* dataset processed per Rubachev et al. [24]. Utility is tracked by (Neg)MSE; fairness by the CvM coefficient (and, where relevant, group-based summaries). Extended discussion is referred to Appendix F.

**Results.** We (i) visualize the effect of $\lambda$ on the CvM–utility frontier, and (ii) study the role of the *smoothness controller* $\varepsilon$ in the soft-ranking operator (iii) demonstrate how adding our CvM-based regularizer translates to decreases in group-based fairness metrics such as DP.

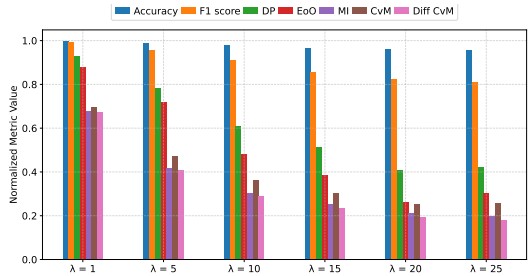

Figure 1: Adult dataset: Plot of utility (accuracy and F1) and fairness (DP, EoO, CvM, MI, Differentiable CvM) metrics. Values of *fair* MLPs ($\lambda \in [1, 5, 10, 15, 20, 25]$, with $L_2$ penalty) normalized by the values of the *unfair* MLP ($\lambda = 0$).

- $\lambda, \varepsilon$-*tuning.* CvM decreases predictably as $\lambda$ increases, exposing a Pareto-like frontier against (Neg)MSE (see Figure 2 and 11). Within practical ranges, $\varepsilon$ exhibits negligible impact on both utility and CvM in this setting (see discussion in Appendix F).

- *Impact of $\lambda$ on DP:* We observe a clear relation between decreasing CvM and lower DP. Since often regulatory policies for fairness in ML focuses on DP, this provides a strategy for determining an appropriate value of $\lambda$ via the reasoning chain "regulations $\rightarrow$ DP $\rightarrow$ $\xi_n \rightarrow \lambda$" , bridging the gap between regulation and the training of deep models.

### 3.3 Hyperparameter strategy

Based on our experiments on the weather forecasting dataset, we propose a scalable and dataset-agnostic hyperparameter tuning pipeline consisting of 3 steps:

**Step 1 (utility-only baseline).** Tune non-fairness hyperparameters with $\lambda = 0$ (architecture, optimizer, regularization,...) to confirm task learnability and provide an initial performance baseline.

**Step 2 (fairness-specific tuning).** Fix the baseline hyperparameters that maximize utility, then sweep the CvM multiplier $\lambda$ and the smoothness controller $\varepsilon$ via randomized search over wide ranges. This provides an initial reference point that enables to shrink down to the regions

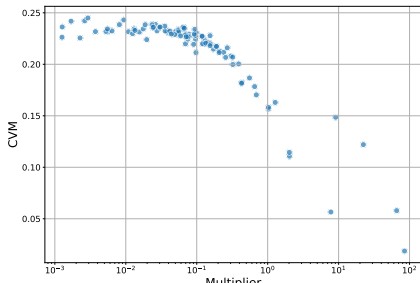

Figure 2: Weather dataset: CvM $\xi(S, \hat{Y})$ vs. regularization multiplier $\lambda$. Increasing $\lambda$ reduces $\xi(S, \hat{Y})$.

of $\lambda$ and $\epsilon$ that are most promising to perform hyperparameter search in step 3.

**Step 3 (penalized-utility selection).** Guide the exploration of hyperparameter space based on a fairness-penalized utility $U(l, c, ; \gamma)$ . In our experiments we define it as:

$$U(\ell, c; \gamma) = \begin{cases} \ell, & c \le \gamma, \\ \ell - \alpha(c - \gamma), & c > \gamma, \end{cases}$$

where $\ell$ is the utility (to maximize), $c$ is the CvM (to minimize), $\gamma$ is a cutoff, and $\alpha$ is a penalty slope. This methodology applies directly to other large tabular datasets beyond weather, and it concentrates the search on desirable regions of the Pareto frontier. The value of the cutoff $\gamma$ should be determined to accomplish the desired levels of group fairness as discussed in F.2

### 3.4 Discussion

With our experiments we (i) demonstrate controllability of the trade-off via the multiplier $\lambda$, (ii) assess robustness and scalability of the differentiable CvM term on large tabular data, (iii) provide a minibatch-friendly tuning methodology suitable for modern optimizers, and (iv) show strong correlation between DP and CvM providing a reference to determine $\lambda$, bridging a gap between DP-based AI regulations and in-training practices.

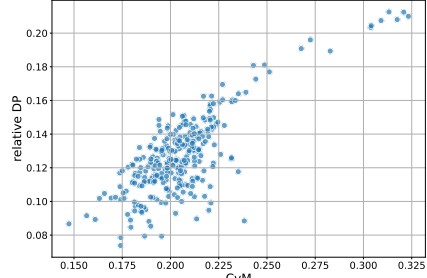

Figure 3: Weather dataset (`sun_elevation` binned): relationship between CvM and the relative DP gap across $\lambda$; lower CvM aligns with smaller DP disparities.

## 4 Conclusion and future work

This work introduces a CvM-based regularizer that makes fairness controllable both during training and as a model fine-tuning method. The CvM term is computed via a finite-sample, rank-based estimator made trainable by replacing non-differentiable ranks with a smooth, order-preserving soft ranking which yields stable gradients that backpropagate efficiently in modern ML training regimes. The approach exposes a single training-time control $\lambda$ for positioning models along the fairness–utility frontier. We also observe a consistent alignment between CvM reductions and reduction in demographic-parity gaps, providing a direct connection between DP-focused policy and training-time decisions via the CvM regularization. Overall, the method offers a lightweight, scalable mechanism to control fairness within modern ML training regimes offering

a practical path for deploying models that are both accurate and equitable even when the available data is imperfect. Future work will extend experiments in the same datasets and to fairness-specific large-scale datasets as well as develop stronger experimental and theoretical connections between CvM and established fairness metrics such as DP and EO.

## 5 Acknowledgments

Albert Gimò received support from "La Caixa" Foundation (ID 100010434) fellowship No LCF/BQ/EU24/12060099.

Federico Pavone received funding from the European Union's Horizon 2020 research and innovation program under the Marie Skłodowska-Curie grant agreement No 101034255.

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

# A Related works

## A.1 Datasets

Fairness evaluation and method design have over-relied on tiny, aging benchmarks (e.g., Adult, COMPAS). We explicitly lean into larger tabular settings: alongside standard Adult experiments, we scale to a processed Weather benchmark from TabReD [24] to emulate real training loads and hyperparameter search, precisely because fairness-aware, open, large tabular datasets are scarce. Using this non-fairness-specific but sizable corpus stresses training throughput and stability in ways small datasets cannot.

FairJob [27] targets fairness in online systems and offers a valuable, real-world dataset and highlights the lack of open-source large fairness-aware datasets. Our present study focuses on supervised tabular tasks with standard deep-learning training loops and widely adopted metrics/APIs (Fairlearn [6]). Aligning protocols and baselines across online/interactive settings is non-trivial and would require additional engineering (ranking/replay, exposure bias control) beyond scope for the workshop draft; we therefore leave a Fairjob-style recsys evaluation to future work.

## A.2 Methods

Fairness in ML is a rapidly evolving field, with mitigation strategies broadly categorized into pre-processing, in-processing, and post-processing methods. This work focuses on the in-processing paradigm, where the fairness objective is integrated directly into the model training process. Our approach, which uses the CvM statistic for regularization, sits at the intersection of two key research areas: dependence-based fairness methods and the use of integral probability metrics in ML.

### A.2.1 In-processing fairness methods

In-processing methods modify the learning algorithm to enforce fairness constraints or penalties during training [23, 30]. One of the most prominent approaches is adversarial debiasing, where a model's representation is trained to be predictive of the target label while simultaneously being unable to predict the sensitive attribute. This is often achieved by training an adversary network to predict the sensitive attribute from the model's latent representation [15, 33]. Another common approach is constrained optimization, which formulates the fairness objective as a constraint on the model's predictions. These methods often use convex optimization techniques to satisfy fairness criteria like demographic parity or equalized odds [31, 32].

Our method differs from these approaches by framing fairness as a direct independence objective between the model's output and the sensitive attribute, and achieving this through a novel regularizer derived from a statistical test, rather than an adversarial game or a hard constraint.

### A.2.2 Dependence-based fairness regularization

A large body of work has sought to achieve fairness by minimizing the statistical dependence between the model's predictions $\hat{Y}$ and the sensitive attribute S. This is often achieved by adding a regularization term to the standard loss function with a penalty as a measure of dependence. Classic dependence measures used for this purpose include mutual information (MI), which quantifies the information shared between two variables [18, 19]. Other work has explored maximal correlation [22] and measures of covariance [9].

More recently, research has leveraged kernel-based independence measures, which can capture non-linear dependencies. The Hilbert-Schmidt Independence Criterion (HSIC) and Maximum Mean Discrepancy (MMD) are two prominent examples. HSIC is a powerful non-parametric measure of dependence that has been widely used in ML for feature selection and independent component analysis [16]. In the context of fairness, it can be used to regularize a model to make its representations independent of the sensitive attribute [25]. MMD is an integral probability metric that measures the distance between two probability distributions and has also been applied to fairness, particularly in fair representation learning [17, 20].

Our work contributes to this line of research by proposing a novel dependence regularizer based on the CvM statistic, a classic goodness-of-fit test. While similar in spirit to MMD as an integral

probability metric, the CvM statistic has distinct properties and provides a new perspective on measuring distributional discrepancy for fairness applications.

### A.2.3 The CvM statistic in ML

The CvM statistic is a well-established tool in classical statistics used to test the goodness-of-fit of a sample's empirical distribution to a given reference distribution [11, 28]. Its use in ML has been more limited but has appeared in contexts such as evaluating user simulations in dialogue systems [29] or as a general-purpose distance for hyperparameter tuning [8].

To the best of our knowledge, the application of the CvM statistic as a direct regularizer for achieving independence-based fairness is a novel contribution. Unlike HSIC and MMD which are based on kernel inner products, the CvM statistic directly compares the cumulative distribution functions (CDFs) of the model outputs across different sensitive groups. This provides a different theoretical foundation and may offer computational or statistical advantages in certain settings.

## B  The CvM dependence coefficient

The CvM coefficient has appeared numerous times in the literature, including [13, 14]. In this appendix we provide an explanation of the CvM dependence coefficient for the purposes of motivating its use for dependence measuring and fairness.

### B.1  Derivation

**Assumptions.**  We assume $Y$ is continuous and that measurability/integrability conditions hold, so that changes of integration order (Fubini/Tonelli) are valid. All distribution functions are right-continuous and non-decreasing.

The coefficient takes inspiration of the CvM distance. Given CDFs $F$ and $G$ on $\mathbb{R}$, the CvM distance is defined as

$$d_{\mathrm{CvM}}^2(F, G) \;=\; \int_{\mathbb{R}} \big(F(t) - G(t)\big)^2 \, dG(t). \tag{3}$$

Measuring the discrepancy between the conditional and marginal laws of $Y$ leads to the dependence functional

$$\xi(X, Y) \;:=\; \int_{\mathcal{X}} \int_{\mathbb{R}} \big(F_{Y|X}(t \mid x) - F_Y(t)\big)^2 \, dF_Y(t) \, dF_X(x). \tag{4}$$

which equals zero iff $F_{Y|X}(\cdot \mid x) = F_Y(\cdot)$ $F_X$-a.s., i.e., when $X$ and $Y$ are independent [13].

This coefficient, which we refer to as the CvM coefficient, allows for a variance-based formulation. Let $p_t(X) := \mathbb{E}[\mathbb{1}_{\{Y \geq t\}} \mid X]$. For continuous $Y$, $p_t(X) = 1 - F_{Y|X}(t \mid X)$ and $\mathbb{E}[p_t(X)] = \mathbb{P}(Y \geq t) = 1 - F_Y(t)$. Expanding the square in (4) and using $\mathbb{E}[F_{Y|X}(t \mid X)] = F_Y(t)$ yields

$$\xi(X, Y) \;=\; \int_{\mathbb{R}} \mathrm{Var}\big(p_t(X)\big) \, dF_Y(t) \;=\; \int_{\mathbb{R}} \mathrm{Var}\Big(\mathbb{E}[\mathbb{1}_{\{Y \geq t\}} \mid X]\Big) \, dF_Y(t). \tag{5}$$

Normalizing by the unconditional variability of the threshold indicators we redefine the coefficeint

$$\xi(X, Y) \;:=\; \frac{\displaystyle\int_{\mathbb{R}} \mathrm{Var}\Big(\mathbb{E}[\mathbb{1}_{\{Y \geq t\}} \mid X]\Big) \, dF_Y(t)}{\displaystyle\int_{\mathbb{R}} \mathrm{Var}\big(\mathbb{1}_{\{Y \geq t\}}\big) \, dF_Y(t)}, \tag{6}$$

which is a form more commonly used in the literature. By the law of total variance applied to $\mathbb{1}_{\{Y \geq t\}}$, the numerator is bounded above by the denominator for every $t$, hence $0 \leq \xi(X, Y) \leq 1$.

If $Y$ is continuous, then with $u = F_Y(t)$ and $U := F_Y(Y) \sim \mathrm{Unif}(0, 1)$,

$$\int_{\mathbb{R}} \mathrm{Var}\big(\mathbb{1}_{\{Y \geq t\}}\big) \, dF_Y(t) = \int_{\mathbb{R}} \big(1 - F_Y(t)\big) F_Y(t) \, dF_Y(t) = \mathbb{E}\big[U(1 - U)\big] = \frac{1}{6}. \tag{7}$$

Thus, for continuous $Y$, the normalizing term in (6) is a constant $1/6$.

## B.2 Interpreting the CvM Coefficient

In this section we aim to provide an intuitive explanation of what is being measured by the CvM coefficient. We provide intuitions both from a variance explanation point of view and from a perspective of measuring differences between distributions. We maintain the continuity assumptions from the previous subsection: $Y$ is continuous, measurability/integrability conditions hold, and changes of integration order are valid.

Consider the normalized dependence coefficient defined in (6). Fix a threshold $t \in \mathbb{R}$ and define the Bernoulli variable $\mathbb{1}_{\{Y \geq t\}}$ with mean $p_Y(t) := \mathbb{P}(Y \geq t)$, and its conditional counterpart $p_{Y|x}(t) := \mathbb{P}(Y \geq t \mid X = x)$. When $X$ is informative about $Y$, the function $x \mapsto p_{Y|x}(t)$ varies across $x$; when $X$ and $Y$ are nearly independent, this variation is small. The numerator in (6) aggregates this signal as

$$\int \mathrm{Var}\big(\mathbb{E}[\mathbb{1}_{\{Y \geq t\}} \mid X]\big) \, dF_Y(t) \;=\; \int \mathrm{Var}\big(p_{Y|X}(t)\big) \, dF_Y(t),$$

so larger values indicate that $X$ explains more of the thresholded behavior of $Y$ across many $t$.

**Illustrative example**

Consider the simple function $Y = \sin(\pi X)$ and an added noise component $\mathcal{N}(0, \sigma^2)$ for different values of $\sigma$ as shown in 4. Fix $t = 0$ for illustration purposes.

Then $x \mapsto \mathbb{P}(Y \geq 0 \mid X = x)$ is close to $\{0, 1\}$ for most $x$ when $\sigma$ is small, becomes less extreme but still variable for moderate $\sigma$, and is almost constant at $1/2$ for large $\sigma$. This differences mean that $\mathrm{Var}(p_{Y|X}(t))$ decreases with $\sigma$, lowering the CvM numerator. As discussed, the denominator stays constant at $\frac{1}{6}$ for all values of $\sigma$. The heatmap over $t$ in Figure 5 shows the same pattern persists across thresholds; integrating over $t$ therefore preserves this effect, leading to lower noise levels being associated to higher CvM values.

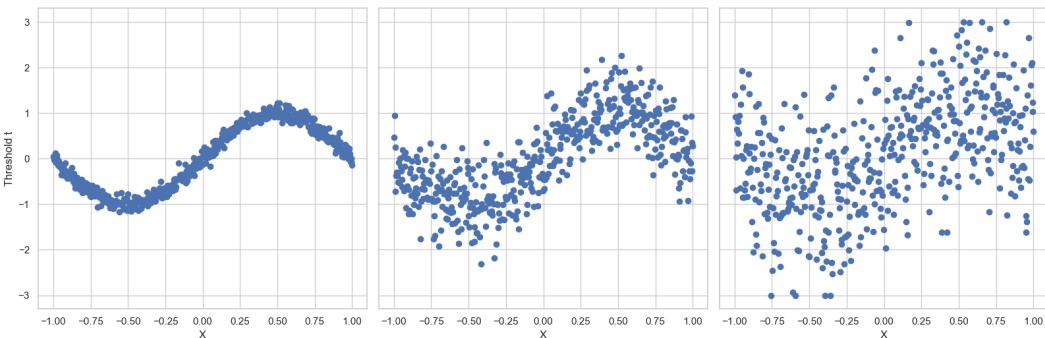

Figure 4: Function $Y = \sin(\pi x) + \mathcal{N}(0, \sigma^2)$ for different $\sigma$ values.

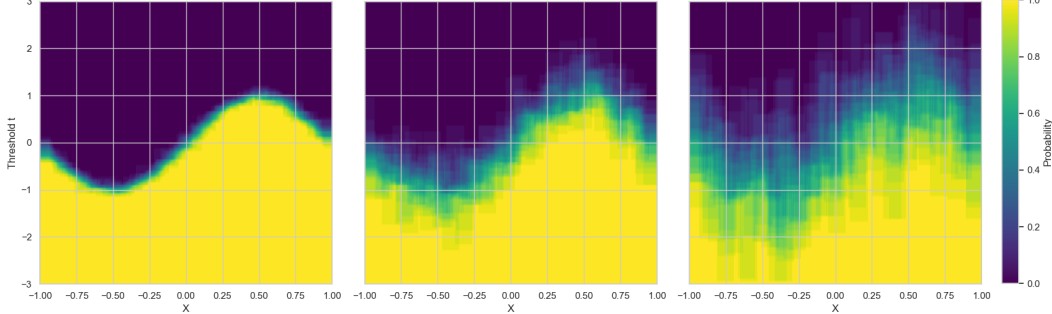

Figure 5: Heatmap encoding the values of $\mathbb{P}(Y \geq t \mid X = x)$ for different cutoffs $t$ (in the Y axes) and different noise levels.

Expanding on this variance-based explanation, by the law of total variance applied to $\mathbb{1}_{\{Y \geq t\}}$, the numerator in (6) measures the component of variance *explained* by $X$, while the denominator captures

the *total* variance of the threshold indicators. Thus, the coefficient admits the interpretation

$$\xi(X,Y) = \frac{\int_{\mathbb{R}} \text{variability in } \mathbb{1}_{\{Y \geq t\}} \text{ explained by } X}{\int_{\mathbb{R}} \text{total variability of } \mathbb{1}_{\{Y \geq t\}}} .$$

The coefficient allows an alternative interpretation. Going back to (4), one can view the numerator as the average (over $x$) squared difference between the conditional CDF $F_{Y|X}(\cdot \mid x)$ and the marginal $F_Y(\cdot)$, integrated over the different values of $X$. This viewpoint emphasizes that $\xi(X,Y)$ is large when conditioning on $X$ substantially deforms the distribution of $Y$, and small when $F_{Y|X}$ remains close to $F_Y$ for most $x$.

### B.3 Connection to group-fairness metrics

Many group-fairness metrics allow an independence based interpretation. The expectation being that enforcing some type of (conditional) independence between the sensitive attribute and the model's predictions will prevent the sensitive attribute from having a disproportionate effect on the outputs of the model.

Let $S$ be a sensitive attribute and $\hat{Y}$ a (continuous) model output. The independence target of demographic parity (DP), $\hat{Y} \perp S$, is satisfied when

$$\xi(S,\hat{Y}) := \frac{\int \text{Var}\Big(\mathbb{E}[\mathbb{1}_{\{\hat{Y} \geq t\}} \mid S]\Big) dF_{\hat{Y}}(t)}{\int \text{Var}\big(\mathbb{1}_{\{\hat{Y} \geq t\}}\big) dF_{\hat{Y}}(t)} = 0 . \tag{8}$$

Equalized odds (EO), which requires $\hat{Y} \perp S \mid Y$, can be addressed by applying the same construction within each outcome stratum (i.e., replacing $F_{\hat{Y}}$ with $F_{\hat{Y}|Y=y}$ and averaging over $y$). We adopt (8) as a differentiable penalty during training; practical estimators are discussed in appendix D.

## C Properties of the estimation $\xi_n$

In this section we explore some properties of the sample-based estimation of the CvM coefficient $\xi_n$. An important result is discussed in the following subsection and relates to the connection between the theoretical and sample-based coefficients. Continuity properties in terms of robustness to perturbations are also discussed.

### C.1 Asymptotic Consistency

The following theorem, presented in Chatterjee [10] provides asymptotic guarantees on the asymptotic accuracy of the estimation:

**Proposition 2** (Theorem 1 in Chatterjee [10]). *If $Y$ is not almost surely constant, as $n \to \infty$,*

$$\xi_n(X,Y) \to \xi(X,Y) := \frac{\int \text{Var}\Big(\mathbb{E}[\mathbb{1}_{\{Y \geq t\}} \mid X]\Big) dF_Y(t)}{\int \text{Var}\big(\mathbb{1}_{\{Y \geq t\}}\big) dF_Y(t)} \in [0,1] .$$

A proof is provided in Chatterjee [10].

### C.2 Robustness to perturbations (continuity)

We use the estimator presented in Chatterjee [10]:

$$\xi_n = 1 - \frac{n \sum_{k=1}^{n-1} |r_{k+1} - r_k|}{2 \sum_{i=1}^{n} l_i (n - l_i)}, \qquad r_i := \#\{ j : x_j \leq x_i \}, \quad l_i := \#\{ j : y_j \geq y_i \}, \tag{9}$$

i.e., $\xi_n$ depends only on the relative orderings of $\{x_i\}$ and $\{y_i\}$ (max-ranks), not on their magnitudes.

**Proposition 3** (Robustness to perturbations restated). *Let $(X_n, Y_n)$ be $n$ i.i.d. samples from $p(X, Y)$. Let $Y$ be continuous and let $Z_n^1, Z_n^2$ contain $n$ i.i.d. samples from a continuous real-valued noise variable. Define $X^\eta := X + \eta Z^1$ and $Y^\eta := Y + \eta Z^2$ for some $\eta$. Then, with probability 1,*

$$\lim_{\eta \to 0} \xi_n(X, Y^\eta) = \xi_n(X, Y),$$

$$\lim_{\eta \to 0} \mathbb{E}\big[\xi_n(X^\eta, Y)\big] = \lim_{\eta \to 0} \mathbb{E}\big[\xi_n(X^\eta, Y^\eta)\big] = \mathbb{E}\big[\xi_n(X, Y)\big], \tag{10}$$

*where the expectations are with respect to the perturbation noise. If $X$ is also continuous, the expectations can be removed.*

*Proof.* Fix a realized sample $(x_1, \ldots, x_n, y_1, \ldots, y_n)$.

*(i) Perturbing $Y$ only.* Since $Y$ is continuous, with probability 1 there are no ties among $\{y_i\}$ and the minimum spacing $\Delta_Y := \min_{i \neq j} |y_i - y_j|$ is strictly positive. Because $Z^2$ takes finite values, there exists some $\eta_0 > 0$ such that for all $\eta \in [0, \eta_0)$ we have $\max_i |\eta Z_i^2| < \Delta_Y/2$, hence the ordering of $\{y_i\}$ is unchanged. The ranks $l_i$ and $r_k$ are also unchanged. By Equation 9, $\xi_n(X, Y^\eta) = \xi_n(X, Y)$ for all sufficiently small $\eta$, yielding $\lim_{\eta \to 0} \xi_n(X, Y^\eta) = \xi_n(X, Y)$ almost surely.

*(ii) Perturbing $X$ (and optionally $Y$).* If $X$ is continuous, the same spacing argument applies to $\{x_i\}$, so for all sufficiently small $\eta$ the $X$-order is unchanged and hence $\xi_n(X^\eta, Y) = \xi_n(X, Y)$ and $\xi_n(X^\eta, Y^\eta) = \xi_n(X, Y)$ almost surely.

If $X$ may have ties, adding arbitrarily small continuous noise acts as a random tie-breaker within each tied block, producing—conditionally on the untied values—the same distribution over strict total orders as uniform random tie-breaking. Taking expectations over the perturbation therefore averages $\xi_n$ over all consistent tie-breakings; this equals the corresponding (noise-free) expectation of $\xi_n(X, Y)$ computed with random tie-breaking. Hence

$$\lim_{\eta \to 0} \mathbb{E}\big[\xi_n(X^\eta, Y)\big] = \mathbb{E}\big[\xi_n(X, Y)\big], \qquad \lim_{\eta \to 0} \mathbb{E}\big[\xi_n(X^\eta, Y^\eta)\big] = \mathbb{E}\big[\xi_n(X, Y)\big].$$

Combining (i) and (ii) proves the claim. $\square$

**Implications.** For continuous variables, $\xi_n$ is insensitive to infinitesimal perturbations, which supports stable training when used as a regularizer. When $X$ is discrete, randomized (or noise-induced) tie-breaking preserves $\xi_n$ in expectation, providing robustness at the level of average behavior.

## D   Implementation of the differential CvM coefficient

Let $r_\Psi^\varepsilon(\theta)$ denote the soft (differentiable) ranking operator defined via projections onto the permutahedron, where $\varepsilon > 0$ is the *smoothness controller* that trades faithfulness to hard ranks for smoother Jacobians [7]. Small $\varepsilon$ yields near-exact ranks but poorly conditioned/less informative derivatives; large $\varepsilon$ produces well-behaved gradients but compresses the dynamic range of the ranks.

*Note:* The smoothness operator $\varepsilon$ is referred to as *regularization strength* in the paper introducing this soft ranking method. We chose to modify this naming to avoid confusion with the multiplier of the CvM regularization $\lambda$.

To showcase the role of this parameter we examine two toy inputs: (i) $n = 15{,}000$ i.i.d. $\mathcal{N}(0, 1)$ samples and (ii) $n = 2{,}000$ equally spaced points on $[0, 1]$ (shuffled). We compare the hard ranks (NumPy) with $r_\Psi^\varepsilon$ for several $\varepsilon$ and sort both outputs for visualization. Perfect agreement would lie on the $45°$ line. As shown in Figure 6, larger $\varepsilon$ preserves order but visibly shrinks the rank spread.

**Order-preserving shrinkage and a simple fix**

Because $r_\Psi^\varepsilon$ is isotonic (order-preserving), the main distortion at larger $\varepsilon$ is magnitude shrinkage rather than order mistakes. We therefore post-process the soft ranks with a monotone affine rescaling to match the endpoints of the true rank range. Let $s \in \mathbb{R}^n$ be the soft ranks for a vector, and $t \in \mathbb{R}^n$ its hard ranks. Define

$$s_{\min} = \min_i s_i, \quad s_{\max} = \max_i s_i, \qquad t_{\min} = \min_i t_i, \quad t_{\max} = \max_i t_i,$$

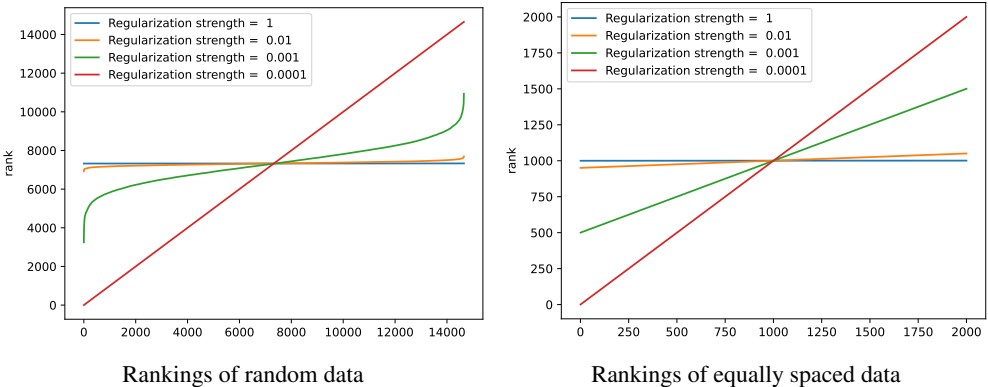

Figure 6: Effect of soft-ranking smoothness $\varepsilon$ on ranks: larger $\varepsilon$ preserves order but shrinks rank spread (Left: 15,000 i.i.d. $\mathcal{N}(0,1)$. Right: 2,000 points in $[0,1]$).

and apply the mapping

$$\tilde{s}_i \;=\; m(s_i) \;:=\; \frac{s_i - s_{\min}}{s_{\max} - s_{\min}}\,(t_{\max} - t_{\min}) + t_{\min}. \tag{11}$$

This rescaling is strictly increasing, preserves the ordering, and matches boundary values ($\tilde{s}_{\arg\min s} = t_{\min}$, $\tilde{s}_{\arg\max s} = t_{\max}$). Its Jacobian with respect to $s$ is a constant scalar factor $(t_{\max} - t_{\min})/(s_{\max} - s_{\min})$, so gradients remain informative and are merely scaled, which improves numerical conditioning without altering the rank-based structure.

Figure 7 shows that the affine correction restores near-linear alignment to the hard ranks for large-$\varepsilon$ soft ranks while retaining smooth derivatives. In all experiments where differentiability is required, we compute $r_\Psi^\varepsilon$ with a moderately large $\varepsilon$ and then apply (11) before using the ranks inside the differentiable $\xi$ computation. When training models we suggest adjusting the value of $\varepsilon$ to the dataset and tuning it as part of the hyperparameter optimization process.

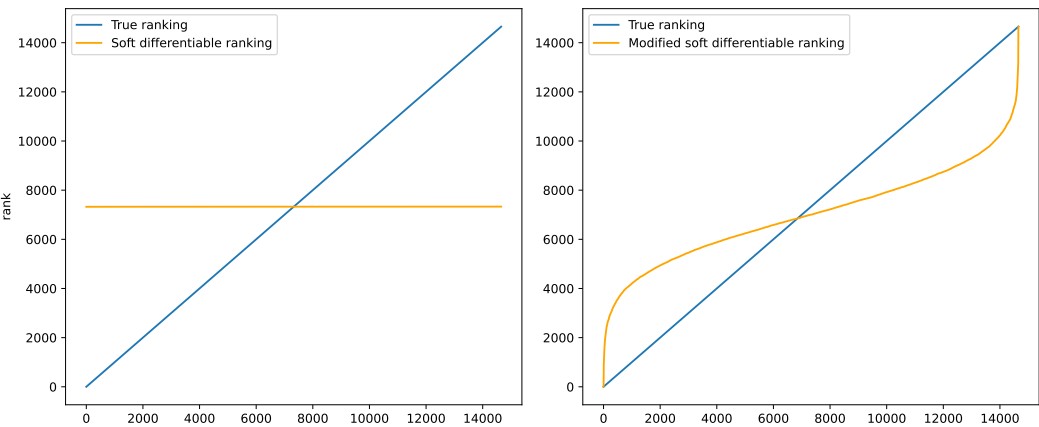

Figure 7: Affine rescaling in Equation 11 restores alignment between soft and hard ranks under large $\varepsilon$ (before vs. after).

# E    Adult dataset: extended results and analysis

**Setup.** We analyze binary income prediction on *Adult* with two sensitive attributes of different informativeness for the label: *gender* (weakly correlated) and *education* (strongly correlated). Models are 3-layer MLPs with layers of sizes [64, 32, 16]; utility is tracked by accuracy/F1, and fairness by (i) CvM and MI (ii) group gaps for DP/EO (max differences across groups). The displayed results

correspond to the averages and statistics computed on 10 independent runs for each value of the multiplier.

The CvM regularizer's effectiveness varies when switching sensitive attributes, especially when the attribute has imbalanced groups. For instance, when using "education" as the sensitive attribute, groups with low counts (e.g., "Preschool" and "Doctorate") were merged into broader categories, which led to more representative unfairness metrics. The reported results are based on this modified education variable. "Preschool", "1st-4th", "5th-6th", "7th-8th", and "9th" were merged into "Less than HS", while "Doctorate" and "Prof-school" were grouped as "High-income Edu", and the rest as their original categories.

**Multiplier effects and correlation regime.** Increasing $\lambda$ lowers CvM and typically shrinks DP/EO gaps, revealing a fairness–utility frontier whose steepness depends on the attribute–label correlation. With *gender*, moderate $\lambda$ achieves noticeable fairness gains with modest accuracy cost; with *education*, fairness improvements incur sharper utility drops, reflecting a harder trade-off. At $\lambda = 0$, models attain peak utility but exhibit higher dependence; as $\lambda$ grows, both unfairness and, eventually, performance decrease. Specific values for the metrics and percentual changes can be observed in Tables 1 and 2, respectively. The results corresponding to Table 1 can be visualized with the corresponding error bars in Figure 8.

| | Utility (maximize) | | Unfairness (minimize) | | | | |
|---|---|---|---|---|---|---|---|
| | Accuracy ($\uparrow$) | F1 score ($\uparrow$) | DP ($\downarrow$) | EoO ($\downarrow$) | CvM ($\downarrow$) | MI ($\downarrow$) | Diff CvM ($\downarrow$) |
| $\lambda = 0$ | **0.8539** | **0.6665** | 0.7868 | 0.7424 | 0.2051 | 0.1468 | 0.1469 |
| $\lambda = 1$ | 0.8530 | 0.6616 | 0.7304 | 0.6534 | 0.1428 | 0.0997 | 0.0990 |
| $\lambda = 5$ | 0.8450 | 0.6391 | 0.6153 | 0.5325 | 0.0974 | 0.0614 | 0.0599 |
| $\lambda = 10$ | 0.8359 | 0.6069 | 0.4797 | 0.3568 | 0.0749 | 0.0444 | 0.0427 |
| $\lambda = 15$ | 0.8261 | 0.5717 | 0.4059 | 0.2855 | 0.0622 | 0.0373 | 0.0349 |
| $\lambda = 20$ | 0.8213 | 0.5494 | **0.3205** | **0.1965** | 0.0516 | **0.0315** | 0.0288 |
| $\lambda = 25$ | 0.8177 | 0.5400 | 0.3308 | 0.2269 | **0.0291** | 0.0530 | **0.0262** |

Table 1: Utility (higher is better) and fairness (lower is better) metrics across regularization strengths $\lambda$. Best values are reported in bold.

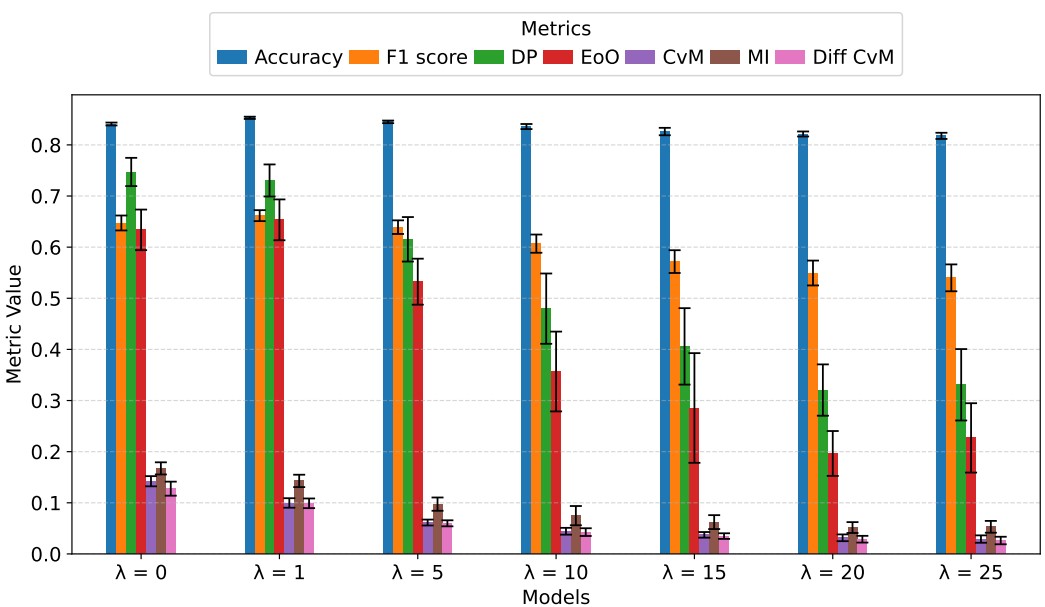

Figure 8: Adult dataset utility (accuracy and F1) and fairness (DP, EoO, CvM, MI, Differentiable CvM) metrics. Values of *unfair* MLP ($\lambda = 0$) and *fair* MLPs ($\lambda \in [1, 5, 10, 15, 20, 25]$, with $L_2$ penalty). Error bars indicate standard deviation computed over 10 runs.

|  | Utility (maximize) | | Unfairness (minimize) | | | | |
|---|---|---|---|---|---|---|---|
|  | Accuracy ($\uparrow$) | F1 score ($\uparrow$) | DP ($\downarrow$) | EoO ($\downarrow$) | CvM ($\downarrow$) | MI ($\downarrow$) | Diff CvM ($\downarrow$) |
| $\lambda = 1$ | **-0.10%** | **-0.73%** | -7.17% | -11.99% | -30.36% | -32.09% | -32.63% |
| $\lambda = 5$ | -1.04% | -4.11% | -21.80% | -28.28% | -52.50% | -58.18% | -59.23% |
| $\lambda = 10$ | -2.11% | -8.94% | -39.04% | -51.94% | -63.48% | -69.76% | -70.94% |
| $\lambda = 15$ | -3.25% | -14.22% | -48.41% | -61.55% | -69.67% | -74.59% | -76.25% |
| $\lambda = 20$ | -3.82% | -17.57% | **-59.27%** | **-73.53%** | -74.84% | **-78.55%** | -80.40% |
| $\lambda = 25$ | -4.24% | -18.98% | -57.96% | -69.44% | **-80.18%** | -74.15% | **-82.17%** |

Table 2: Relative metrics change with respect to the baseline ($\lambda = 0$) across regularization strengths $\lambda$. Best values are reported in bold.

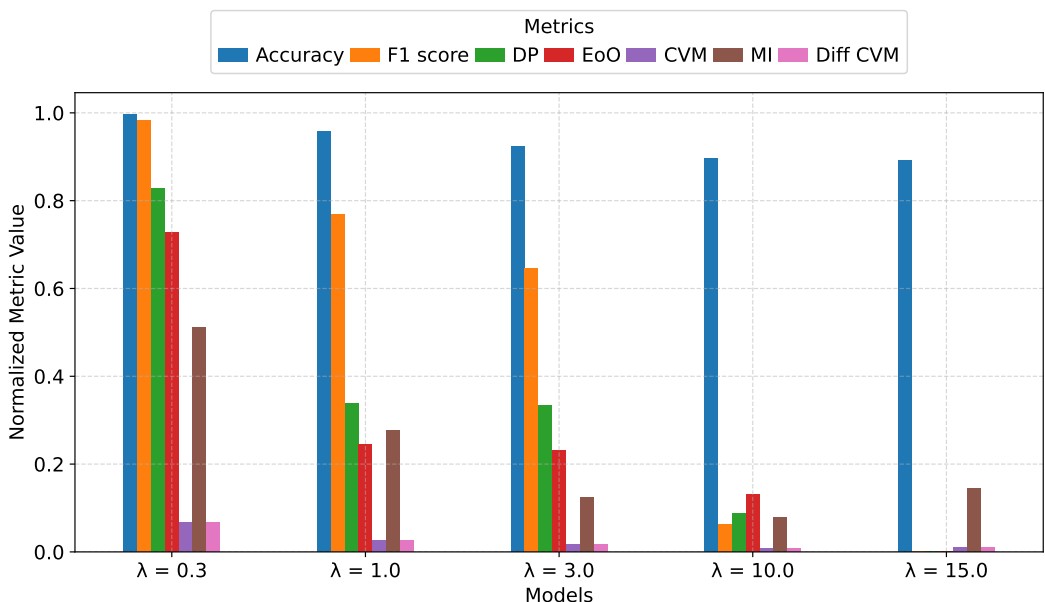

Figure 9: Utility (accuracy, F1 score) and fairness (DP, EO, CvM, MI, Differentiable CvM) metrics comparison of *regularized* models ($\lambda \in [1, 3, 10, 15]$, with $L_1$ penalty) normalized by values of *unfair* model ($\lambda = 0$). Compared to $L_2$ regularization, we observe increased instability when training with $L_1$ loss for the regularizer.

**L₁ vs. L₂ on the CvM term.** As observed in Figure 9, the increased sensitivity on the $\lambda$ parameter leads to models failing to learn the task and assigning almost all the labels to the majority class as observed for $\lambda \in \{10, 15\}$ in the plot. Even for small values of the multiplier (compared to those used in Figure 1) the decrease in performance is considerable. As observed in Figure 9, in some cases when the value of the multiplier is set too high (see $\lambda = 15$) the regularizer term can take over and the model fail to learn the task. A practical method to avoid this phenomenon is to use $L_2$ regularization instead of $L_1$. Using $L_2$ regularization makes the magnitude of the corresponding gradient be proportional to the value of the CvM coefficient. More precisely, given a coefficient $\alpha$ for $L_1$ regularization and $\beta$ for $L_2$ regularization then the regularization strength is higher for $L_1$ if $|\xi_n| < \frac{\alpha}{2\beta}$ and stronger for $L_2$ if $|\xi_n| > \frac{\alpha}{2\beta}$. Intuitively, $L_2$ penalizes small $\xi$ more gently when its value is close to 0 (which prevents collapse when $\lambda$ is large), while increasing pressure on clearly unfair solutions as dependence increases. Empirically, $L_2$ yields more stable training and preserves utility more consistently than $L_1$ for comparable fairness gains (Figure 1, Figure 9). We therefore adopt the following $L_2$ formulation for subsequent runs:

$$\theta^* = \text{argmin}_{\theta \in \Theta}\ \mathcal{L}(\theta) + \lambda\, \xi^2(S, Y_\theta). \quad (12)$$

**Fine-tuning schedule.** We also explore the possibility of using the regularizer for fine-tuning. We start by training the model with no regularization and from that unregularized checkpoint progressively

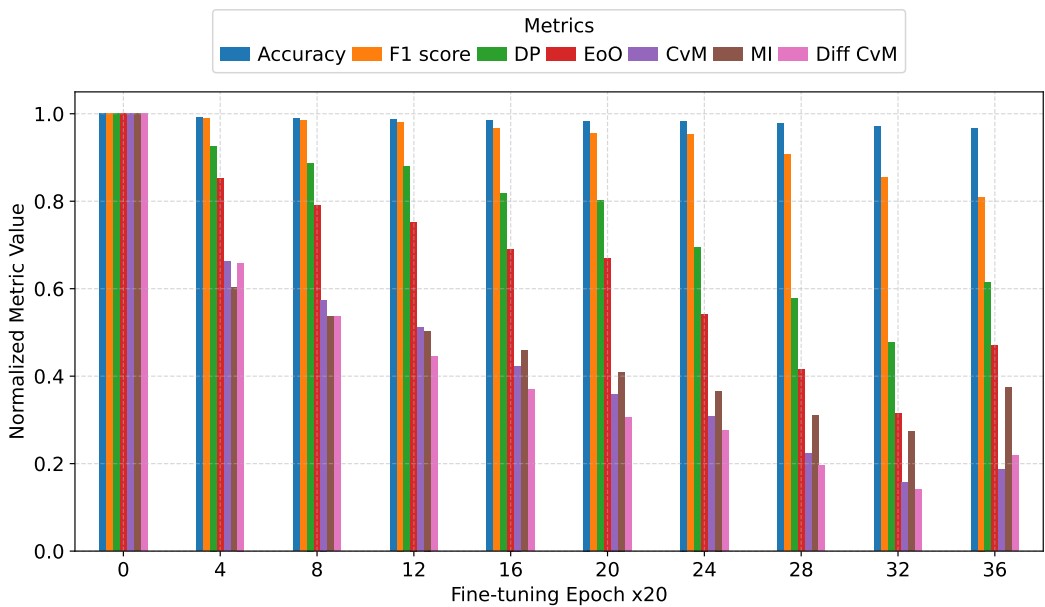

Figure 10: Utility (accuracy, F1 score) and fairness (DP, EO, CvM, MI, Differentiable CvM) metrics comparison normalized by values of *unfair* model ($\lambda = 0$). The unregularized model is trained for 20 epochs and then the value of $\lambda$ is geometrically increased every 80 epochs. It helps control the stability of the optimization model and not crush into trivial solutions

increase the CvM multiplier to trace a controlled path toward fairness. The process is the following: We train 20 epochs with $\lambda = 0$, then introduce the CvM term and increase $\lambda$ by $\sqrt{3}$ every 80 epochs. This schedule steadily reduces CvM and DP/EO gaps while maintaining competitive accuracy/F1, avoiding the abrupt utility losses seen when starting with a large $\lambda$.

*Remark:* Finite-sample rank-based estimates can be slightly negative under near-independence. This is expected from sampling variability and is mitigated in practice by using continuous outputs (probabilities) and an $L_2$ penalty on the CvM term.

**Takeaways.** On *Adult*, the CvM regularizer enables calibrated movement along the fairness–utility frontier; the effect is gentle for weakly correlated attributes and steeper for strongly correlated ones. $L_2$ regularization and a staged fine-tuning schedule improve robustness, yielding smoother progress toward lower dependence (CvM) and smaller DP/EO gaps with controlled utility cost.

# F  Weather forecasting dataset: extended results and analysis

We study large-scale temperature prediction using the *Weather Forecasting* dataset processed per Rubachev et al. [24]. Utility is tracked by (Neg)MSE; fairness by the CvM dependence coefficient and MI. When the sensitive attribute is discrete fairness is also tracked by DP. Because the dataset lacks informative categorical attributes, group structure is obtained by discretizing a continuous variable via binning into subsets of equal size.

## F.1  Hyperparameter tuning to control the fairness–performance trade-off

**Tuning targets.** We tune (i) the CvM multiplier $\lambda$ and (ii) the derivative smoothness controller $\varepsilon$ of the soft-ranking operator. The initial $\lambda$ sweep spans $[10^{-3}, 10^2]$; subsequent sweeps adapt to $[10^{-1}, 10^3]$ based on observed frontiers. For $\varepsilon$, we start with an initial wide sweep were no effect is observed. After we reduce the search space to a narrow, practically stable band $[10^{-5}, 10^{-3}]$ to prioritize budget on $\lambda$.

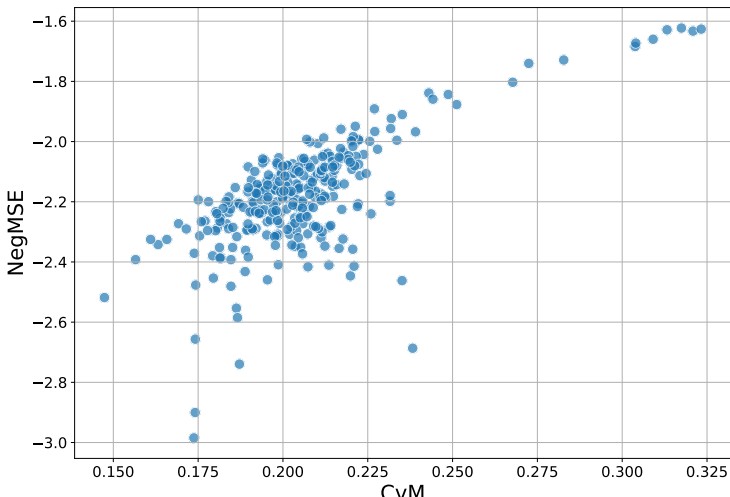

Figure 11: Weather dataset: CvM $\xi(S, \hat{Y})$ vs. NegMSE. This plot enables the visualization of the fairness-utility frontier.

**Optimization protocol.** We use randomized search with Optuna [3, 26], maximizing a reported objective (see below) with early stopping (patience 16). Budgets: $N=300$ trials when using a discretized (binned) sensitive attribute for group metrics, and $N=100$ trials when treating the sensitive signal as continuous (CvM-only). We use the default train/val/test splits from the TabReD preprocessing and take `sun_elevation` as the sensitive attribute, which is strongly correlated with the target ($\approx 0.47$).

### F.1.1 Fairness-penalized hyperparameter selection

For generating the results, we follow the same method that we detail in the paper. Namely we conduct the following 3 steps.

**Step 1 (utility-only baseline).** Tune non-fairness hyperparameters with $\lambda=0$ (architecture, optimizer, regularization, early stopping) to establish a performance baseline and confirm task learnability.

**Step 2 (fairness-specific tuning).** Fix the hyperparameters from the previous step that maximize utility, then sweep the CvM multiplier $\lambda$ and the smoothness controller $\varepsilon$ via randomized search over wide ranges. For this specific case, we identify that $\varepsilon$ has a negligible effect on the results so we keep its range limited to $[10^{-6}, 10^{-3}]$ and concentrate most of the exploration power to explore the effect of $\lambda \in [10^{-1}, 10^3]$. This initial search over the regularizer-specific hyperparameters provides an initial reference point that enables us to shrink down to the regions of $\lambda$ and $\epsilon$ that are most promising for hyperparameter search in the following step.

**Step 3 (penalized-utility selection).** Report to Optuna a fairness-penalized utility score that preserves utility when CvM is below a cutoff and subtracts a linear penalty otherwise:

$$U(\ell, c; \gamma) = \begin{cases} \ell, & c \leq \gamma, \\ \ell - \alpha\,(c - \gamma), & c > \gamma, \end{cases} \tag{13}$$

where $\ell$ is the utility to maximize (NegMSE), $c$ is the CvM value to minimize, and $\gamma$ is a user-specified cutoff reflecting the desired group-fairness regime (e.g., via the empirical relation between CvM and DP). We fixed $\alpha=10$ and note that other slopes can be explored to adjust selection pressure. This pipeline is dataset-agnostic and directly applicable to other large tabular problems.

### F.2 Choosing the cutoff $\gamma$

The choice of the cutoff $\gamma$ will guide the hyperparameter exploration of the hyperparamter optimization module. As seen in Figure 12, the values of the CvM (determined by the choice of $\gamma$) will center around the cutoff value.

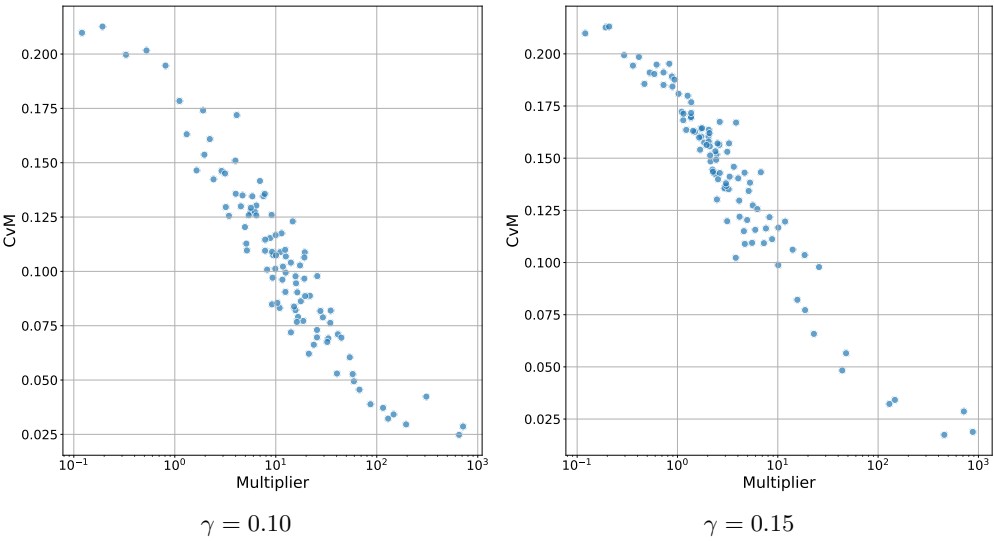

$\gamma = 0.10$         $\gamma = 0.15$

Figure 12: The cutoff determines what values of the CvM runs will be accumulated around and guides the optimization process.

The choice for the value of the cutoff should be task-specific and can be oriented by the relationship between the CvM and other fairness metrics via the discussed relationship between CvM and metrics such as DP. Plots as the one observed in Figure 3 can be useful to guide this choice.

### F.3 Results

We conduct two complementary analyses:

(i) a *continuous* setting, using `sun_elevation` directly for CvM (no groups),

ii) a *discrete* setting, where the same variable is *binned* to form groups for DP evaluation.

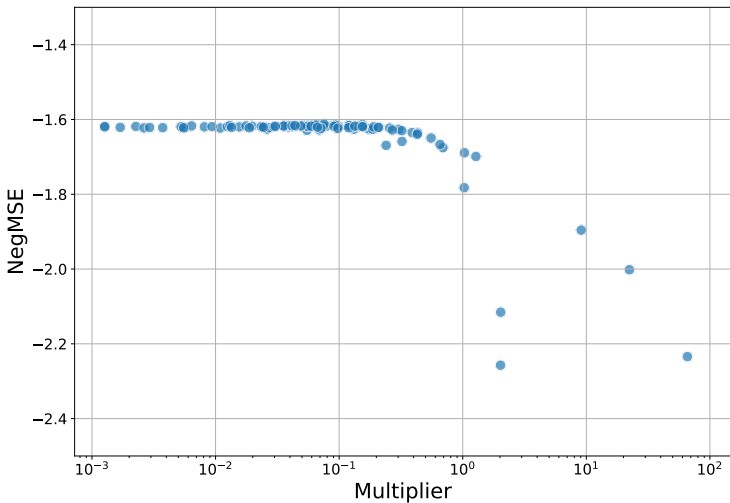

Figure 13: Effect of setting different values of $\lambda$ on the obtained NegMSE values. An increase of the multiplier leads to a deterioration of performance.

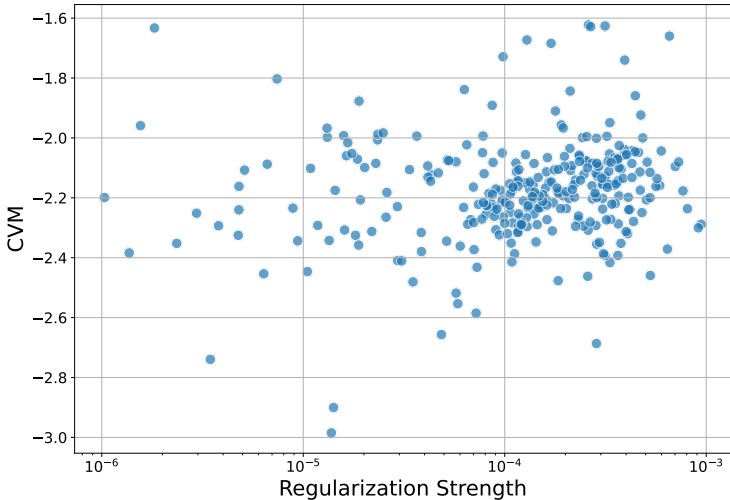

Figure 14: Effect of setting different values of $\varepsilon$ on the obtained CvM values. There is a negligible effect of $\varepsilon$ on the results.

In both settings, increasing $\lambda$ monotonically reduces CvM, yielding a Pareto-like frontier against NegMSE, see Figure 11[2]. Moreover, $\varepsilon$ exerts limited influence on the CvM–utility trade-off relative to $\lambda$, justifying a narrowed search space for $\varepsilon$ (Figure 14). In the binned (discrete) analysis, reductions in CvM are accompanied by consistent decreases in DP gaps, enabling a practical mapping from policy targets on DP to choices of $\lambda$ (Figure 3)[3] with model training.

**Visualization.** We summarize the trade-off via Pareto-like plots with utility on the horizontal axis and fairness (CvM or DP) on the vertical axis as observed in Figure 11; each point corresponds to a distinct training with a different $\lambda$ value. To visualize the impact of the regularization parameter $\lambda$ on the different metrics, Figure 15 depicts the joint evolution of NegMSE, DP, CvM, and MI as $\lambda$ varies. For this analysis, the training runs were sorted by their corresponding $\lambda$ values and divided into eight groups containing an equal number of runs, from which the statistics for each bin were computed.

## G   Additional experimental details

**Resources.** Experiments were conducted on internal cluster on instances with a RAM of 500Go and 46 CPUs available and 2 GPUs V100.

**Practical Guidance**

- Use continuous outputs for stability; report DP/EO alongside CvM.
- Prioritize tuning $\lambda$; treat $\varepsilon$ as a low-priority *derivative smoothness controller* (fix small values unless instability is observed).
- Prefer $L_2$ on the CvM term; consider ramping $\lambda$ for fine-tuning when utility is critical.
- For hyper parameter optimization, adopt a fairness-penalized utility to target the desired region of the frontier.

---

[2]In figures involving NegMSE, two outlier runs were removed for visibility; both corresponded to very large $\lambda$ yielding low CvM and very poor utility.

[3]Demographic parity can be thought of as a stronger version of the US Equal Employment Opportunity Commission's "four-fifths rule", which requires that the "selection rate for any race, sex, or ethnic group [must be at least] four-fifths (4/5) (or eighty percent) of the rate for the group with the highest rate", see the Uniform Guidelines on Employment Selection Procedures, 29 C.F.R. §1607.4(D) (2015).

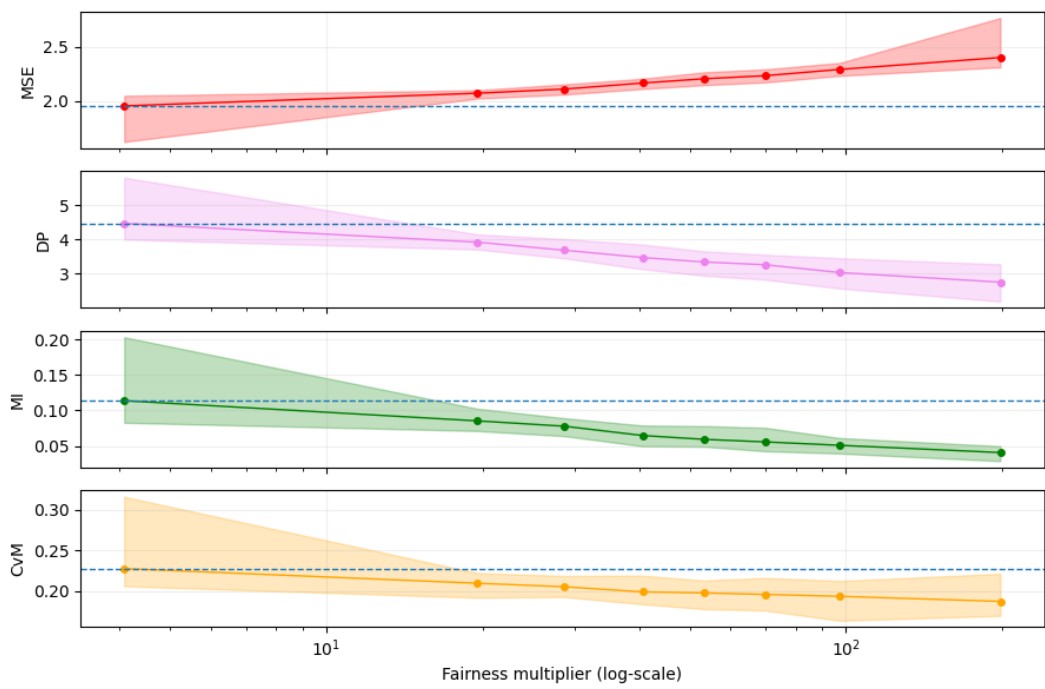

Figure 15: Joint evolution of NegMSE, DP, CvM, and MI as the regularization strength $\lambda$ varies. The plot summarizes results from 300 training runs, which were sorted by their corresponding $\lambda$ values and divided into eight groups with an equal number of runs. Dotted lines indicate the metric values obtained from runs with $\lambda = 0$.

