# OpenReview forum: "Fairness Through Independence via Cramér-von Mises Regularization"
_NeurIPS.cc/2025/Workshop/Reliable_ML — NeurIPS 2025 - Reliable ML Workshop_

### Official Review · Reviewer_K9CR · 2025-09-20
**Interesting and Novel Approach, Lacking Benchmarking**

**Rating:** 7
**Confidence:** 4

**Review:**

**Summary:**
The authors propose a novel in-processing method for controlling fairness in machine learning models. This method involves adding a regularization term to the loss function inspired by a dependence-based statistical metric, the Cramér-von Mises (CvM) coefficient. Specifically, they adapt the metric, mainly using a soft (differentiable) ranking algorithm, to integrate it well into stochastic optimization. Then, they explore the tradeoff between utility and fairness by varying the multiplier of the CvM regularization. They also suggest a method for using their regularizer approach in fine-tuning. Lastly, they present a workflow via penalized-utility selection to manage the aforementioned trade-off in large tabular workloads and also establish a connection between CvM coefficient and demographic parity (DP) scores.

**Strengths:**
1. The use of the CvM statistic in the fairness literature seems to be an interesting and novel direction.
2. The paper is generally easy to follow and well organized.
3. The paper falls within the scope of the workshop.

**Weaknesses / Limitations:**
1. Missing direct experimental comparison with previous work (e.g., other dependence-based regularization methods).
2. Experimental setup (number of runs,  etc.) is not very clear, especially for the Adult dataset.

**Suggestions:**
1. It would be interesting to achieve some deeper theoretical understanding of the connection of CvM and fairness metrics like DP/EO as the authors themselves note.
2. The authors should compare the CvM approach with some other regularizers used in the literature like MMD and explore the performance/fairness tradeoff.
3. Minor mistakes: In line 390 I believe it should read "there exists some (random) $\eta _0$".  In line 447 "cf ??". Lines 456-459 are unclear, F1 score is missing from Figures 8 and 9. In the legend of Figure 11 I believe $\gamma$ instead of $\lambda$ should be used. In the adult dataset, it is unclear how weakly and strongly correlated features are compared. Also I think there is an error in the citation for [33].